# A Pentavalent HIV-1 Subtype C Vaccine Containing Computationally Selected gp120 Strains Improves the Breadth of V1V2 Region Responses

**DOI:** 10.3390/vaccines13020133

**Published:** 2025-01-28

**Authors:** Xiaoying Shen, Bette Korber, Rachel L. Spreng, Sheetal S. Sawant, Allan deCamp, Arthur S. McMillan, Ryan Mathura, Susan Zolla-Pazner, Abraham Pinter, Robert Parks, Cindy Bowman, Laura Sutherland, Richard Scearce, Nicole L. Yates, David C. Montefiori, Barton F. Haynes, Georgia D. Tomaras

**Affiliations:** 1Duke Human Vaccine Institute, Duke University, Durham, NC 27710, USA; rachel.spreng@duke.edu (R.L.S.); sheetal.sawant@duke.edu (S.S.S.); asmcmillan92@gmail.com (A.S.M.); mr.ryanmathura@gmail.com (R.M.); rob.parks@duke.edu (R.P.); cindy.bowman@duke.edu (C.B.); laura.sutherland@duke.edu (L.S.); rscearce@duke.edu (R.S.); nicole.yates@duke.edu (N.L.Y.); david.montefiori@duke.edu (D.C.M.); barton.haynes@duke.edu (B.F.H.); 2Department of Surgery, Duke University, Durham, NC 27710, USA; 3Los Alamos National Laboratory, The New Mexico Consortium, Los Alamos, NM 87544, USA; btk@lanl.gov; 4Vaccine and Infectious Disease Division, Fred Hutchinson Cancer Research Center, Seattle, WA 98109, USA; 5Icahn School of Medicine at Mount Sinai, New York, NY 10029, USA; susan.zolla-pazner@mssm.edu; 6Public Health Research Institute, New Jersey Medical School, Rutgers University, Newark, NJ 07103, USA; pinterab@njms.rutgers.edu; 7Department of Medicine, Duke University School of Medicine, Durham, NC 27710, USA; 8Department of Immunology, Duke University Medical Center, Durham, NC 27710, USA; 9Department of Molecular Genetics and Microbiology, Duke University Medical Center, Durham, NC 27710, USA

**Keywords:** HIV vaccine, antibody response, polyvalent vaccine, breadth, V1V2 response

## Abstract

Background: HIV-1 envelope (Env) variable loops 1 and 2 (V1V2) directed non-neutralizing antibodies were a correlate of decreased transmission risk in the RV144 vaccine trial. Thus, the elicitation and breadth of antibody responses against the V1V2 of HIV-1 Env are important considerations for HIV-1 vaccine candidates. The V1V2 region’s highly variable nature and the extensive diversity of subtype C HIV-1 Envelopes (Envs) make the V1V2 response breadth a high priority for HIV-1 vaccine regimens aiming for V1V2-mediated protection in Southern Africa. Here, we determined whether the breadth of the anti-V1V2 vaccine response can be broadened by including HIV-1 Env strains computationally designed to enhance the coverage of subtype C V1V2 sequence diversity. Methods: Three subtype C Env strains were selected to maximize antibody binding coverage while complementing subtype C vaccine gp120s that were given in human clinical trials in South Africa, as well as to improve epitope accessibility. Humoral immunogenicity of a novel trivalent gp120 vaccine immunogen, a bivalent gp120 boost already in clinical trials (1086C and TV1), and a pentavalent (all five gp120s combined) were evaluated in a preclinical immunization study in guinea pigs. The pentavalent combination was further evaluated with alum versus glucopyranosyl lipid adjuvants formulated in squalene-in-water emulsion (GLA-SE) adjuvants in non-human primates. The breadth of the anti-V1V2 response was assessed using an array of cross-subtype variable loops 1&2 (V1V2) scaffold proteins and linear V2 peptides. Results: The breadth of the IgG response against V1V2 antigens of the trivalent and pentavalent groups was comparable, and both were greater than the breadth of the bivalent group. Linear epitope mapping showed that two linear epitopes in V2 were targeted by the vaccinated animals: the V2 hotspot focused at ^169^K that potentially correlated with decreased HIV-1 risk in RV144 and the V2.2 site (^179^LDV/I^181^) that is part of the integrin α4β7 binding site. The bivalent vaccine elicited a significantly higher magnitude of binding to the V2 hotspot compared to the trivalent vaccine whereas the trivalent vaccine elicited significantly higher binding to the V2.2 epitope compared to the bivalent vaccine, while the pentavalent recognized both regions. Conclusions: These results demonstrate that the three new computationally selected subtype C Envs successfully complemented 1086C and TV1 for broader V1V2 antibody responses, and, in concert with adjuvants that stimulate V1V2 responses, can be considered as part of a rationale immunogen design to improve V1V2 IgG coverage in future vaccine trials in South Africa.

## 1. Introduction

Antibody responses against the V2 region of HIV-1 Env are of high interest for HIV-1 vaccine development. Analysis of the RV144 Thai trial showed plasma IgG anti-V1V2 responses correlated with decreased risk of HIV-1 transmission [1,2,3]. A sieve analysis that compared the V1V2 sequences of breakthrough viruses from participants who received either active vaccines or the placebo in RV144 identified two signatures in V2 at amino acid (aa) positions 169 and 181 (defined relative to the HXB2 reference strain), suggesting the involvement of vaccine-elicited immune responses targeting these V2 regions in the partial protection observed in the study [4]. Additional immunological studies of RV144 and non-human primate (NHP) preclinical studies shed light on potential mechanisms through which antibody responses targeting V2 regions including those encompassing aa169 and aa181 could contribute to vaccine efficacy [5,6,7].

First, an RV144 linear epitope mapping study identified a hotspot linear epitope in V2 containing aa169 and binding to the epitope correlated with lower risk of infection [8,9]. A class of V2-targeting monoclonal antibodies which require lysine at 169 (K169) for binding were later isolated from RV144 vaccines [10]. These antibodies can mediate antibody-dependent cell-mediated cytotoxicity (ADCC), virion capture, and antibody-dependent cellular phagocytosis suggesting Fc-mediated function as a potential mechanism of function of these antibodies [10,11,12].

With 7.7 million people living with HIV and 150,000 new infections in 2023, South Africa (SA) continues to have the largest AIDS epidemic in the world [13]. The Pox-Protein Public Private Partnership (P5) program was established to develop a subtype C-directed vaccine based on the RV144 regimen [14]. One challenge faced by any contemporary vaccine candidate for SA and elsewhere is that HIV-1 is steadily diversifying. The dominant subtype in Thailand is the circulating recombinant form CRF01, believed to be a subtype A and E recombinant. CRF01 likely spread from Africa where it had been more diverse, seeding the Thai epidemic around 1987, and it began diversifying as an Asian sub-lineage from a common ancestor in that time frame [15]. The RV144 trial began in 2003, ~20 years later. In contrast, the C subtype, the dominant HIV-1 subtype in SA, has a significantly older origin, estimated to be in the mid-1960s [16]; thus, recent trials conducted in South Africa have to contend with the diversity that has accumulated within subtype C over a 50-year time span, far more extensive than that of CRF01 at the time the RV144 trial was conducted [17] (Appendix A). The genetic and antigenic diversity within subtypes, and so globally, continues to expand over time [17,18]. P5 identified three subtype C candidate Env strains for testing the RV144-like poxvirus prime, protein boost vaccine regimens in SA. Phylogenetic analysis revealed a 21–23% average amino acid (AA) distance between the three P5 subtype C strains (C.96ZM651, 1086C, and TV1) [19] and the circulating subtype C stains in SA (Figure 1A), compared to the 16–17% average AA distance between the RV144 Thai trial CRF01_AE vaccine strains and the circulating CRF01_AE stains in Thailand (Appendix A). Furthermore, immunogenicity data obtained from two P5 clinical trials in SA, testing either the subtype B/E RV144 vaccine (HVTN 097 [20]) or a subtype C vaccine that followed the vaccine delivery strategy of RV144 (HVTN 100 [21]), revealed a relatively lower magnitude and breadth of the V1V2 response from the subtype C vaccine compared to the CRF01_AE vaccine [22]. The phase 2b/3 HVTN 702 testing of the same subtype C vaccine regimen as that used in HVTN 100 in South Africa showed no evidence of efficacy [23]. Two other recently completed efficacy trials testing viral vector (Ad26) and clade C Env vaccine regimens also failed to show efficacy [24,25]. These findings further highlight the need for identifying better strategies such as additional Env immunogen strains to elicit an increased breadth of V2 responses against circulating subtype C viruses.

Several human and NHP vaccine studies have triggered the development of V1V2 responses with breadth. V1V2-targeting antibodies that cross-react with multiple subgroups and recognize both conformational and linear epitopes were identified in samples from RV144 vaccines [26]. Cross-reactive V1V2 antibodies have also been found in HIV-1 infected individuals [27,28,29] and have been elicited by vaccination of NHPs [5].

An NHP study comparing bivalent subtypes BE versus pentavalent subtypes BEEE boosts reported that the inclusion of additional diverse gp120 immunogens to a pox-prime/protein boost regimen can augment antibody responses and enhance protection against a simian–human immunodeficiency virus (SHIV) challenge [30]. A study using V1V2-scaffold proteins further showed the advantage of multivalent V1V2 immunization in improving heterologous V1V2-specific responses compared to monovalent immunizations [5]. We hypothesized that the great diversity of subtype C Env necessitated an immunogen design with the capacity for eliciting more diverse V1V2 responses. Considering the importance of the potentially protective V1V2 immune correlates in the RV144 Thai trial, we aimed to improve the breadth of vaccine-elicited responses against SA subtype C strains by including natural Env variants that were computationally selected to enhance the V1V2-region coverage of P5 subtype C vaccine strains already in use, 1086C and TV1. The three novel subtype C Env strains we chose were tested as a trivalent combination in guinea pigs, side by side with the 1086C and TV1 bivalent regimen and pentavalent regimen that included all five subtype C strains. We evaluated the breadth of binding responses to both V1V2 scaffolded proteins and V2 linear epitopes, and both in terms of subtype/strain coverage and epitope coverage. Figure 1Diversity of subtype C HIV-1 Env and vaccine coverage. (**A**) An unrooted phylogenetic tree of 995 subtype C Env complete sequences available through the Los Alamos HIV database web alignment, www.hiv.lanl.gov, circa 2016 when this vaccine design was originally undertaken. South Africa, a key nation for vaccine trials and the focus of this study, is compared to C clade viruses globally, with two distinctive subtype C clusters originating in India and Brazil highlighted. This tree was generated with FastTree [31] and the figure made with Rainbow Tree (www.hiv.lanl.gov). (**B**) The net charge distribution of the combined V1 h and V2 h hypervariable regions across the acute subtype C sequences included in (**A**). The original P5 vaccine V1 h and V2 h hypervariable regions were all relatively negatively charged (−3), while the trivalent selections were neutral or positive. (**C**) V1 h and V2 h length distributions. Two of the tree original vaccines had very long V1 h and V2 h regions; the three complementary strains selected ranged from short to average. (**D**) LOGOS showing the amino acid frequencies in each position between Env 153–184. The relative size of each amino acid reflects the proportion of the variant in the virus population as indicated (C acute, C global, and Not-C). The top figure shows the amino acid frequencies found in the pentavalent vaccine, with the bivalent coverage indicated in black, and the augmented diversity coverage of the trivalent vaccine shown in blue. The next three figures show the diversity coverage of the vaccine in gray for the bivalent, blue for the augmented coverage of the trivalent, and red for the missed amino acids in the pentavalent combinations. The most common missed amino acids would have been covered by the inclusion of ZM651, which was the assumed prime vaccine strain in the initial design. (**E**) Sequence alignment of the V2 epitope region and properties across the combined V1 and V2 hypervariable regions for the subtype C vaccine candidates.
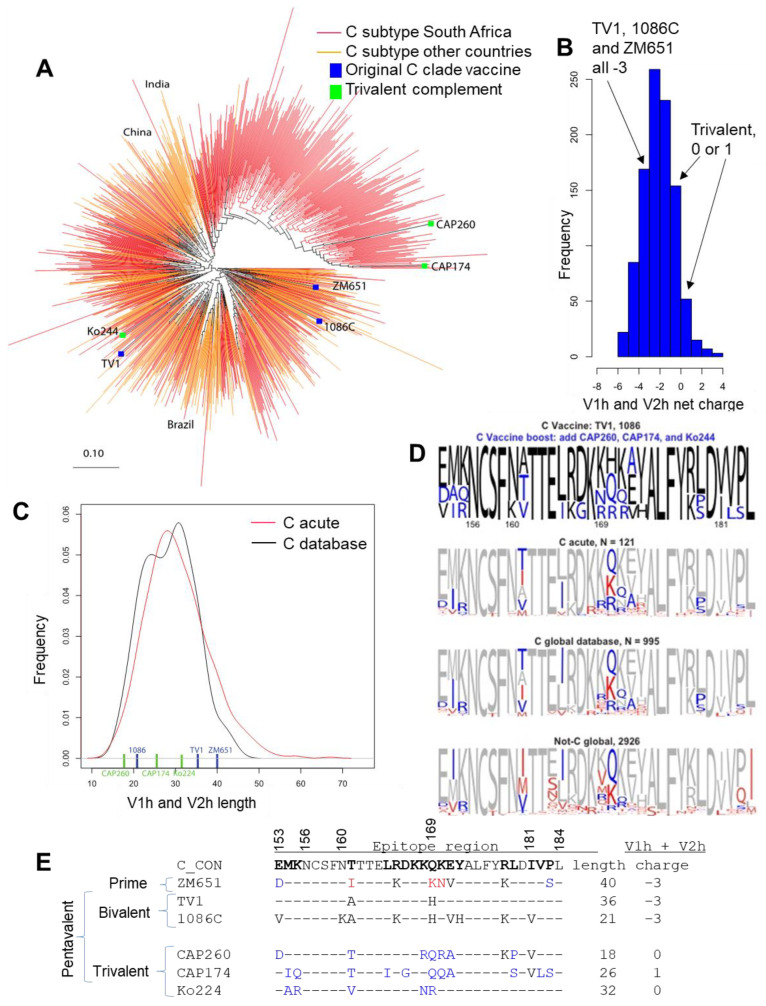


The overall breadth of the cross-subtype V1V2 binding response was comparable between the trivalent and the pentavalent groups, both being higher than that of the bivalent group. We also found that the aluminum-adjuvanted group had improved or non-inferior V1V2 responses, as tested by three different complimentary methods, compared to the GLA-SE group in the non-human primate study. The pentavalent subtype C regimen with the aluminum adjuvant elicited the best breadth for V1V2 binding and provided the best coverage among the three vaccine regimens of the two contiguous linear V2 epitopes that covered the two sites of immune pressure identified in the RV144 sieve analysis.

## 2. Materials and Methods

### 2.1. Env Subunit Proteins for Vaccination

Gp120 recombinant proteins used for the immunization of guinea pigs were produced by the Duke Human Vaccine Institute Protein Production Facility, Duke University Medical Center, as previously described [32]. Sequences for 1086C and TV1 are previously published [33]. Sequences for the three newly selected strains are available in GenBank as CAP260: JN681228, CAP174: JN967791, and Ko224: JN681240.

### 2.2. Immunization of Guinea Pigs and Cynomolgus Macaques

Guinea pigs for the study were maintained in accordance with the National Institutes of Health and Duke University guidelines and all studies were approved by the appropriate Institutional Animal Care and Use Committee. Guinea pigs in the study were female, Hartley outbred strain from Charles Reiver, and were 3–4 months of age at the time of the first immunization. A total of 3 groups of 6 animals each were immunized intramuscularly with either CAP174 + CAP260 + Ko224 gp120 proteins (trivalent), 1086C + TV1 gp120 proteins (bivalent), or CAP174 + CAP260 + Ko224 + 1086C + TV1 (pentavalent). Each vaccine dose contained 100 μg per animal per protein and was supplied in 15% STR8S-C (STS + R848 + oCpGs) [34] for a total volume of 400 μL. Each dose was split in halves and injected into 1 hindlimb and 1 forelimb. All animals were immunized on week 0, 3, 6, and 9 of the study for a total of 4 immunizations.

Mauritian cynomolgus macaques for the NHP study were housed at BioQUAL, Inc. (Rockville, MD, USA), and all studies were approved by both Duke and BioQUAL animal protocol review committees. BioQUAL is fully accredited by the Association for the Assessment and Accreditation of Laboratory Animal Care International (AAALAC). A total of 6 adult female macaques and 2 adult male macaques were split into 2 groups with 1 male per group. Each macaque received 500 ug total Env protein (100 ug each Env), adjuvanted with either alum/rehydrogel or GLA-SE (25 μg) in a total of 1 mL and split into 0.5 mL per side (bilateral quadriceps).

### 2.3. HIV-1 Specific Binding Antibody Assay

HIV-1 specific IgG antibodies to gp120/gp140 proteins, Env peptides, and V1/V2 scaffolds were measured by an HIV-1 BAMA as previously described [2,35], with the exception that the detecting antibody was biotinylated goat anti-guinea pig IgG (Jackson ImmunoResearch, West Grove, PA, USA) for the guinea pig samples and biotinylated goat anti-monkey IgG (Jackson ImmunoResearch, PA, USA) for the macaque samples. Antibody measurements were acquired on a Bio-Plex instrument (Bio-Rad, Hercules, CA, USA) and the readout was in MFI. The half maximal effective concentration (EC_50_) was calculated using the drc R package [36]. The positivity of the plasma binding response was determined with the following positivity criteria: (1) MFI > antigen-specific cutoff (which is the 95th percentile MFI of all baseline plasma samples, and 100 minimum); (2) MFI > 3-fold MFI of the matched baseline sample.

### 2.4. Peptide Microarray Linear Epitope Mapping

Serum epitope mapping of heterologous strains was performed as previously described [9,37] with minor modifications. Briefly, array slides were provided by JPT Peptide Technologies GmbH (Berlin, Germany) by printing a library designed by Dr. B. Korber, Los Alamos National Laboratory, onto Epoxy glass slides (PolyAn GmbH, Berlin, Germany). The library contains overlapping peptides (15 mers overlapping by 12) covering 7 full length gp160 consensus sequences (subtype A, B, C, D, Group M, CRF1, and CRF2 consensuses) and gp120 sequences of 6 vaccine strains (MN, A244, Th023, C.TV1, ZM641, 1086C). Three identical subarrays, each containing the full peptide library, were printed on each slide. Array slides were scanned at a wavelength of 635 nm with an InnoScan 710 AL scanner (Innopsys, Carbonne, France) using XDR mode. Images were analyzed using MagPix 8.0 software to obtain binding intensity values for all peptides. Linear epitope-specific binding rectal IgG responses were assessed by peptide array as previously described [9]. The magnitude of binding was calculated as the log_2_ fold difference, post-/pre-immunization intensity using the R pepStat package (version 1.41.0). 

### 2.5. Monoclonal Antibody Competition ELISAs

Monoclonal antibody competition assays were performed as described previously [38]. In brief, plates were coated with HIV-1 envelope protein, washed, and blocked. Guinea pig serum samples were diluted 1 to 50 and incubated in triplicate wells for 90 min, followed by the addition of biotinylated monoclonal antibodies at sub-saturating concentrations and a 1 h incubation. The same antibody, in non-biotinylated form, was used to block itself as a positive control. An anti-influenza antibody CH65 was included as a negative control. The binding of biotinylated monoclonal antibodies was determined with horseradish peroxidase (HRP)-conjugated streptavidin. The binding of the biotinylated monoclonal antibody to the HIV-1 envelope in the presence and absence of the competing antibody was compared to calculate the percent inhibition of binding. Historical data on the negative controls of the assay were tracked, and an assay was considered valid if the positive control antibodies blocked greater than 20% of the biotinylated antibody binding.

### 2.6. Magnitude–Breadth Score Calculation

The magnitude–breadth of responses was evaluated for binding to V1V2 antigens and V2 linear epitopes. A magnitude–breadth score was calculated for each animal as the area under the magnitude–breadth curve (AUC), where the magnitude–breadth curve represents the proportion of V1V2 antigens or overlapping V2 peptides for which the animal had a positive vaccine-induced response at each magnitude interval.

### 2.7. Statistical Analyses

SAS software [Version 9.4 of the SAS System Copyright © 2002–2012 by SAS Institute Inc., Cary, NC, USA] was used for performing statistical comparisons. The exact Wilcoxon rank sum test was used for comparison of BAMA EC50 or intensities of binding to epitopes among groups in the guinea pig study. All tests were 2-sided, and *p*-values < 0.05 were considered significant. Due to the exploratory nature of this study, *p*-values were not corrected for multiple comparisons. No statistical comparisons were performed for the NHP study due to the small group size of the study.

## 3. Results

### 3.1. Selection of Novel Subtype C Env for Complimentary V1V2 Coverage

We computationally selected three Env sequences to complement the three P5 subtype C vaccine strains for coverage of subtype C V1V2 sequence diversity. Previous studies found that longer combined hypervariable regions within the V1 and V2 loops (V1V2) are associated with resistance to V2 Apex antibody neutralization [38]. Virus sensitivity for broadly neutralizing antibodies that target V1V2 quaternary epitopes is associated with glycosylation at specific residues, the shorter hypervariable region length in the V1 and V2 loops (V1 h and V2 h), and a positive charge in the V2 h region [39,40,41]. Therefore, in addition to striving to enhance coverage of V1V2 epitope diversity in the epitope regions, we also selected for the following: (1) shorter V1 and V2 loop hypervariable region loop lengths; (2) positive net charge in the V1 h and V2 h regions; and (3) maintenance of conserved glycosylation sites (Figure 1). The combination of vaccine antigens used in the HVTN702 trial had viruses that tended to be unfavorable in terms of these attributes, and to have limited coverage of regional diversity (Figure 1). We also focused on the use of Envs sampled during acute infection for vaccine selection.

To make our selections, we first retrieved 121 subtype C sequences isolated from individuals sampled in SA during acute infection from the LANL database, and then we determined the combined length and net charge of the hypervariable regions in the V1 and V2 loops using the hypervariable region characteristics tool at the Los Alamos HIV database (https://www.hiv.lanl.gov/content/sequence/VAR_REG_CHAR/index.html, accessed on 2 March 2017). A subset of 17 sequences were down-selected that had favorable V1V2 region characteristics, a net charge > 0, and total hypervariable region length < 32, as we hypothesized these would enable better accessibility to the V1V2 epitope region (Figure 1). Next, we selected the three sequences among these that provide the best complementary coverage of the V2 region to the existing subtype C vaccine strains (ZM651, TV1, and 1086C), such that the combination of the six variants enabled us to cover all the common amino acid variants in each position in the V2 epitope region (Figure 1). The three sequences selected were CAP260 (JN681228), CAP174 (JN967791), and Ko224 (JN681240). We then confirmed that the key glycosylation sites N156 and N160 that are associated with sensitivity to quaternary V1V2 broadly neutralizing antibody sensitivity [41,42] are present in the newly selected strains (Figure 1B), and that they are not extreme outliers of the subtype C (Figure 1A). The subtype C phylogenetic tree in Figure 1A shows where the vaccines are distributed relative to the C subtype globally. About half of the SA sequences form a major and distinctive clade, and none of the three original P5 vaccine strains are a member of that sub-lineage. In contrast, two out of the three newly selected boost strains (CAP260 and CAP174) belong to that distinctive SA clade. Thus, the natural diversity of subtype C Env is better sampled with the three newly selected boost strains to the three original P5 subtype C strains even though the selection of these three novel strains was based solely on the V1V2 region (Figure 1A). In addition, V1V2 sequence analysis showed excellent coverage of all the most common amino acids that were found in SA isolates in each position within the V1V2 epitope (Figure 1D), indicating that restricting the selection to shorter V1V2 length and positive charge did not compromise our ability to capture the coverage of V1V2 natural amino acid diversity. Here, we only study the two boost strains from the P5 vaccine regimen and the newly selected subtype C Envs to optimize the boost Env strains. ZM651, which is the prime vaccine strain in ALVAC in the P5 studies, is not tested in these in the study because this study aimed to test for optimal boost strains. Diversity coverage by ZM651 would be provided by the ALAVAC prime in a prime/boost regimen using the optimal clade C Env protein combination determined by these exploratory studies as the boost vaccine. The impact of ZM651 on the diversity coverage is highlighted in Figure 1.

### 3.2. Pentavalent C Immunogenicity Study in Guinea Pigs

Three groups of six guinea pigs each were immunized with trivalent (the three newly selected boost strains: CAP260, CAP174, and Ko224), bivalent (the two original P5 boost strains: TV1 and 1086C), and pentavalent (trivalent + bivalent) gp120 proteins, delivered four times total on week 0, 3, 6, and 9 in STR8S-C adjuvant (Table 1). Serum samples from 2 weeks post the last immunization were used for immunogenicity evaluation in the study. One animal each in the trivalent and pentavalent group died before the last immunization, therefore leaving five animals each for these two groups for the immunogenicity comparison.

The three new protein boost sequences were initially designed to complement all three original P5 subtype C sequences in both the prime and boost (Figure 1). In this study, we focus on subunit protein immunizations; therefore, the prime Env strain C.96ZM651 is not included here, but still, much of the enhanced diversity coverage is retained. We showed this by examining the amino acid variability in V1V2 with and without the three new booster strains added to the two original P5 boost proteins (1086C and TV1) and found that the addition of these new sequences enabled the improved coverage of most major subtype C amino acid variants that were not covered by the original P5 boost strains (Appendix A).

### 3.3. Differences Between Trivalent and Bivalent Groups for Magnitude of V1V2 Response

The plasma IgG response against the V1V2 region of HIV-1 Env 2 weeks after the last immunization of the guinea pigs was evaluated using binding antibody multiplex assays (BAMA). The antigen panel included twenty-three gp70-V1V2 scaffolds, three of which matched or partially matched to the bivalent vaccine strains. Among the twenty non-vaccine-matched (heterologous) strains, eleven were subtype C, six were subtype B, and the other three scaffolds were of subtype A and CRF01_AE (the subtype and country of origin information is published previously [43]).

BAMA results showed that after four immunizations, guinea pigs in all three vaccine groups developed binding responses against all variants of the V1V2 scaffold tested (Table 2, Appendix A). The trivalent group developed higher response against 12 out of 20 heterologous V1V2 scaffolds (*p* < 0.05, two-tailed Wilcoxon rank sum test) compared to the bivalent group, with the group median EC50 titers four- to forty-four-fold higher than that of the bivalent group (Table 2, Appendix A). In particular, the trivalent group developed higher responses to all six subtype B V1V2 scaffolds in the antigen panel (Appendix A, Table 2). Compared to the trivalent group, the bivalent group developed higher responses against the 1086C vaccine-matched V1V2 scaffold (gp70.Ce1086_B2_V1V2) and against two of the twenty heterologous V1V2 scaffolds (Appendix A, Table 2).

### 3.4. V1V2 Responses in Pentavalent Group

The magnitudes of binding responses of the pentavalent group were generally similar to the highest response between the trivalent and the bivalent groups for the V1V2 scaffolds (Figure 2, Table 2). Particularly, for nine out of the twelve V1V2 scaffolds against which the trivalent group developed significantly higher responses compared to the bivalent group, the pentavalent group also developed responses that were significantly higher than those of the bivalent group with three- to sixteen-fold differences between group median EC50 titers (*p* < 0.01, two-tailed Wilcoxon rank sum test; Table 2). For all these 12 V1V2 scaffolds, the binding responses for the pentavalent group were comparable to those of the trivalent group (*p* > 0.05; Appendix A, Table 2). For one of the two heterologous V1V2 scaffolds against which the bivalent group developed significantly higher responses than the trivalent group, gp70.CM244.ec1_V1V2, the pentavalent group developed a binding response that was higher than that of the trivalent group (*p* < 0.05 and four-fold higher by median EC50; Table 2), albeit still significantly lower than that of the bivalent group (*p* < 0.01; Appendix A, Table 2). For the other V1V2 in which there was a higher response in the bivalent group compared to the trivalent group, the response of the pentavalent group was comparable to that of the bivalent group and trended higher than that of the trivalent group even though the difference (four-fold for group median EC50 titers) was not significant (*p* > 0.05; Appendix A, Table 2).

### 3.5. Improved Breadth of V1V2-IgG Binding Response in Trivalent and Pentavalent Groups Compared to Bivalent Group

The breadth of the V1V2-IgG antibody responses was evaluated for each guinea pig as the median EC50 value against the 20 heterologous V1V2 scaffolds (Figure 3A). Both the trivalent and the pentavalent group showed higher levels of overall V1V2 binding breadth (median EC50 for 20 V1V2 scaffolds per animal) compared to the bivalent group (Wilcoxon rank sum *p* = 0.004 and 0.009 for trivalent and pentavalent, respectively) while the median EC50 values were comparable between the trivalent and the pentavalent groups, indicating better overall breadth of V1V2 binding response for the trivalent and pentavalent groups (Figure 3A). In addition, responses in the bivalent group showed a trend of larger variations among the V1V2 scaffolds compared to those in the other two groups.

The breadth of binding to V1V2 antigens, either to all heterologous V1V2 (20 antigens, Figure 3A) or to heterologous subtype C V1V2 (11 antigens, Figure 3B), was examined. We found that for binding to either just subtype C V1V2 antigens, or for all heterologous V1V2 antigens, the pentavalent group and the trivalent group showed comparable breadth, and both were higher than the bivalent group (*p* = 0.017 or 0.03, Wilcoxon rank sum test).

### 3.6. Binding Responses Against V1V2 and V2 Tags and Env Proteins

V1V2 responses were also evaluated using V1V2 constructs without tgp70 backbones to confirm V1V2 responses observed without the potential complications of scaffolding backbones. V1V2 Tags proteins that contain either the V1V2 or just the V2 region of C.1086 and AE.A244 Env placed between an immunoglobulin (Ig) leader sequence and avidin and His tags [10] were tested against plasma samples in BAMA. Binding to the vaccine-matched 1086C V1V2 Tags was significantly higher for the bivalent group compared to the trivalent group and the pentavalent group, with two-tailed Wilcoxon rank sum test *p* values of 0.004 for both (Table 2, Appendix A). Binding to AE.244 V1V2 Tags was also significantly higher for the bivalent group compared to the trivalent group (*p* = 0.008, two-tailed Wilcoxon rank sum test).

Plasma samples following the fourth immunization were also tested for binding to four Env proteins to see how the different vaccine compositions affected Env-binding responses. There was no significant difference between the trivalent and bivalent groups for binding to four HIV Env gp120 and gp140 proteins including a TV1 gp120, which was included in the bivalent group (Table 2, Appendix A). A significant difference was observed between the pentavalent and the bivalent group for binding to a subtype B consensus gp120, Con.6 gp120_B (*p* = 0.009, two-tailed Wilcoxon rank sum test; Table 2).

### 3.7. Pentavalent Group Targeted Both V2 Hotspot and V2.2 Linear Epitopes

Peptide microarray linear epitope mapping assays were performed to profile the specificity of IgG binding responses elicited in the vaccinated guinea pigs against overlapping HIV-1 Env peptides. Env peptides in the microarray library include overlapping peptides covering gp160 of consensus clade A, B, C, D, group M, CRF-01_AE, and CRF-02_AG; and gp120 of vaccine strains including 1086C and TV1, but none of the three new clade C strains (CAP260, CAP174, and Ko224) that were tested in the current study. The results showed that the binding responses were dominated by V3 and C5 binding responses, consistent with the previous observation in non-human primates after HIV-1 Env immunizations [37], followed by a C2 binding response (Figure 4A). The trivalent and pentavalent groups showed a trend for a higher magnitude of binding to a V5–C5 epitope compared to the bivalent group, and the pentavalent group showed a trend for a higher magnitude of binding to a C1 epitope compared to the trivalent and bivalent groups. Statistical tests were not performed to compare the responses among groups for these epitopes.

Antibody responses against contiguous V2 epitopes were subdominant but present in all three groups (Figure 4A,B). The trivalent group and the bivalent group differed strikingly in the fine specificity of the V2 response. The V2 response in the bivalent group was focused on the contiguous epitope that centers around amino acid K169/peptide #53 (V2.hotspot), the V2 hotspot epitope identified in the RV144 Thai trial (Figure 4B and Appendix A) [9]. In contrast, the binding response to this V2.hotpspot epitope is largely missing in the trivalent group, while the trivalent focused on the contiguous V2 epitope (V2.2) that centers around the purported α4β7 binding motif in V2, LDV/I, in peptide #57&58 (Figure 4B and Appendix A). The pentavalent group, however, developed binding responses to both linear epitopes (Figure 4B).

When the magnitudes of the binding responses to the two linear epitopes were compared among groups, the bivalent and pentavalent groups had significantly higher signal intensities compared to the trivalent group for binding to the V2.hotspot of C.1086 or the strain with the highest V2.hotspot binding, with *p* < 0.01 (two-tailed Wilcoxon rank sum test) for both groups for 1086C (Table 3). For V2.2 binding, the trivalent group was significantly higher compared to the bivalent group for TV1 (*p* = 0.022, two-tailed Wilcoxon rank sum test; Table 3). The pentavalent group developed V2.hotspot binding that was comparable to that of the bivalent group (Figure 4B Table 3), and V2.2 binding that was comparable to that of the trivalent group (Figure 4B, Table 3). Note that the peptide microarray library contains matched sequences for the bivalent vaccine isolates TV1 and 1086C, but no matched sequence for any of the three newly selected clade C strains in the trivalent and pentavalent groups.

### 3.8. Different Strain and Epitope Coverage by Bivalent and Trivalent Immunizations

The bivalent and the trivalent group focused on two different V2 linear epitopes, and responses against the two V2 epitopes were specific for different Env strains. Binding to the V2.hotspot epitope was focused on 1086C and were cross-reactive with the CRF01 Thai variant AE.A244 (as well as AE.Th023) for the bivalent group. The trivalent vaccine responses were mostly against C.TV1, A.con, and B.con, while the pentavalent group showed broader cross-clade V2.hotspot coverage (Figure 5A). The 1086C sequence contributed to 67% and 64% of the total V2.hotspot binding in the bivalent and pentavalent groups, respectively (Appendix A). Binding to the V2.2 epitope, in contrast, was seen mostly in the trivalent and pentavalent groups and was focused on TV1 for both the trivalent and the pentavalent groups (5B), with the TV1 sequence contributing to 99% and 91% of the total V2.2 binding in the trivalent and the pentavalent groups, respectively (Appendix A). The bivalent group did not respond to TV1 V2.2 (Table 3), but bound at low levels to B con, C con, C.1086, C.96ZM651, AE.A244, and AE.Th023 for V2.2 (Figure 5B).

We further evaluated the breadth of linear V2 peptide binding responses as measured in microarray. The magnitude–breadth score was calculated for each animal with the magnitude of binding to all linear V2 peptides in the array library for which any of the three vaccine groups showed a positivity rate > 33.3%. Peptides that meet the criteria and the magnitude of binding to these peptides by each animal are shown in Appendix A. The pentavalent group showed a higher magnitude–breadth of binding to V2 linear peptides compared to the trivalent group (*p* = 0.016, Wilcoxon rank sum test), and trended higher compared to the bivalent group (Figure 5C).

### 3.9. Development of Antibody Specificities That Target Binding Sites of Known Antibodies with Virus Inhibitory Functions

The ability of the trivalent, bivalent, and pentavalent subtype C vaccines to elicit antibody specificities that block the binding of an ADCC-mediating antibody A32 and a V2 linear epitope binding antibody CH58 to Env was evaluated. The bivalent group showed a slightly higher level of blocking activity against A32 binding (1.2-fold difference, *p* = 0.018 post immunization #4, Wilcoxon rank sum test) but a substantially higher level of blocking activity against CH58 binding (3.3-fold difference, *p* = 0.0062) compared to the trivalent group (Figure 6; Table 4). Meanwhile, the pentavalent group showed similar levels (*p* > 0.05) of blocking activity compared to the bivalent group against both monoclonal antibodies (mAbs), and a higher level of blocking than the trivalent group (*p* = 0.009 post immunization #4, Wilcoxon rank sum test). Therefore, the bivalent and the pentavalent groups showed advantages over the trivalent group in eliciting antibody specificities that target CH58 and, to a lesser degree, A32 binding sites.

### 3.10. Pentavalent Subtype E/C Vaccine in Non-Human Primates

The pentavalent C vaccine was further tested in cynomolgus macaques for immunogenicity evaluation with two different protein adjuvants. Since a C.TV1-specific V1V2 response was not strongly elicited in the bivalent or pentavalent groups in the guinea pig study, the C.TV1 strain was substituted with AE.A244 in this E/C pentavalent study in macaques. Two groups of four macaques each were given the pentavalent vaccine, adjuvanted in either alum (rehydrogel) or GLA-SE at study week 0, 4, and 8 (Table 5). Humoral immune responses were evaluated using plasma samples collected at week 10 (2 weeks post the third immunization).

BAMA data for serum samples from 2 weeks post the last immunization showed that all four macaques in the alum group developed V1V2-specific responses against all 22 cross-subtype V1V2 scaffolds in the assay (Figure 7A). Meanwhile, the four macaques in the GLA-SE group also developed positive responses against 16 out of the 22 V1V2 scaffolds. Interestingly, even though the overall binding magnitudes to the gp120 and gp140 Env proteins tested were similar between the two study groups (Figure 7B), the alum group showed a consistent trend of a higher magnitude of binding to V1V2 scaffolds and Tags compared to the GLA-SE group (*p* < 0.05 for seven out of twenty-seven V1V2 analytes) (Figure 7A; Appendix A). The magnitude–breadth of binding to V1V2 scaffolds trended higher for the alum group compared to the GLA-SE group (Figure 8).

Further profiling of contiguous linear epitopes targeted by the vaccinated animals showed that the two groups of macaques targeted the same epitopes across Env (Figure 9A). The magnitudes of binding for epitopes in C1, V3, and V5 were comparable between the two groups or with a relatively higher magnitude in the GLA-SE group compared to the alum group (Figure 9A). Binding to V2.hotspot was a cross-subtype in both groups with subtype/strain coverage similar to that of the guinea pig pentavalent group (Figure 9B). The magnitude of binding to AE.A244 V2.hotspot was relatively higher than to the C.1086C V2.hotspot for both animal groups, reflecting the inclusion of AE.A244 in the pentavalent regimen. No substantial binding to the V2.2 epitope was observed in either macaque group (Figure 9B). Consistent with the BAMA observation of a higher V1V2 scaffold binding magnitude, the binding magnitude for the V2.hotspot linear epitope also trended higher for the alum group compared to the GLA-SE group, especially for the C.1086 strain (Figure 9B). The overall breadth of binding to the contiguous linear epitopes of all strains within the V2 region was comparable between the two vaccine groups (Appendix A).

In addition, the ability of vaccine-elicited serum antibodies to block CH58 and A32 binding to the AE.A244 gp120 protein was measured using longitudinal serum samples. The blocking data revealed that macaques in the alum group developed a higher magnitude of antibody response that blocks CH58 binding compared to the GLA-SE group (*p* = 0.03; Appendix A), whereas the ability to block A32 binding is comparable between the two groups (Figure 9C,D). This is consistent with BAMA data showing comparable binding to gp120 proteins but higher levels of binding to most V1V2 constructs by the alum group compared to the GLA-SE group.

## 4. Discussion

The aim of the study was to test whether the three new subtype C boost strains that were selected computationally for better coverage of V1V2 diversity could improve the breadth of anti-V1V2 antibody responses. Three new subtype C Envs (CAP174, CAP260, and Ko224) were selected to complement the original subtype C P5 strains to maximize the coverage of diversities in the V2 region for South African subtype C isolates and that had the hypervariable region characteristics we reasoned might make the V1V2 epitope region more accessible in vaccination than the original C P5 strains. The immunogenicity of (1) the trivalent vaccine containing the three novel subtype C gp120 strains selected by computational sequence analysis, (2) the bivalent vaccine containing 1086C and TV1, two subtype C vaccine strains that were originally selected as the P5 subtype C boost strains, and (3) the pentavalent vaccine containing all five vaccine strains were tested for immunogenicity in guinea pigs. A pentavalent E/C vaccine, consisting of the four subtype C strains in the pentavalent C vaccine plus AE.A244, was further evaluated in macaques.

We found a substantially higher breadth of V1V2 responses from the trivalent (CAP174, CAP260, and Ko224) immunization compared to the bivalent 1086C and TV1 immunization. The trivalent group showed significantly higher binding to 12 out of the 22 heterologous V1V2 scaffolds compared to the bivalent group (*p* < 0.05, two-tailed Wilcoxon rank sum test; Table 2). In particular, the trivalent group was higher for binding to all six subtype B V1V2 scaffolds tested. The pentavalent vaccine elicited the broadest breadth for binding to the V1V2 scaffold panel.

Another measure of the breadth of V1V2 response is the coverage of responses to different V2 linear epitopes. These vaccines elicited binding responses to two linear V2 epitopes. While the bivalent vaccine elicited binding responses against the hotspot V2 epitope (V2.hotspot) that was also targeted by RV144 vaccines and proven and correlate of protection in the RV144 trial, the trivalent vaccine elicited binding responses targeting the V2 epitope that overlaps with the α4β7 binding motif (V2.2). The pentavalent group elicited responses covering both epitopes, with levels comparable (*p* > 0.05, two-tailed Wilcoxon rank sum test) to those of the bivalent group for V2 hotspot binding and levels comparable to those of the trivalent group (*p* > 0.05) for V2.2 binding. Therefore, data from both BAMA and the peptide microarray showed a decisive advantage of the pentavalent group for the breadth and coverage of anti-V1V2 binding antibody responses.

The distinct biases of the V2 linear epitopes targeted by the trivalent and the bivalent vaccines was unexpected and of high significance. Antibodies against the V2 hotspot epitope have been observed in RV144 [10,44] and other human clinical trials [20]. The binding antibody response targeting the V2.2 epitope that was observed in this study has been reported in another preclinical vaccine study [45]. The aa181 signature identified in the RV144 sieve analysis [4] indicates potential immune pressure on an epitope overlapping with this amino acid. The anti-V2.2 response has been postulated to, and theoretically could, be protective due to its potential to interfere with the gut-homing of HIV-1-infected CD4+ T cells. The finding that certain HIV-1 Envs may specifically or preferentially elicit antibodies targeting either V2.hotspot or the V2.2 linear epitope warrants careful immunogen selection for further vaccine development that aims to elicit V2 responses. The V2 signature of RV144 suggests that Abs to both epitopes were targeted, or that the 181 signature could reflect a structural requirement—with a mutation at I181 resulting in a structural change that occludes K169 [46]. It is likely that some Env strains can effectively elicit responses against both epitopes; otherwise, the coverage of both epitopes, if desired, can be achieved by a combination of Env strains as demonstrated by the pentavalent group in the current study.

We further demonstrated that a similar cross-subtype breadth for binding to the V2 hotspot linear epitope can also be elicited in macaques with a pentavalent E/C vaccine regimen including the three new subtype C vaccine strains, C.1086, and AE.A244. AE.A244 was included in this study instead of TV.1 in the guinea pig study because TV.1 was shown to not significantly contribute to V2.hotspot and V1V2 scaffold binding responses which are potential correlates of protection in RV144, whereas AE.A244 strongly elicited V1V2 responses in RV144 and in the guinea pig study. No substantial binding to the V2.2 linear epitope was observed in either group of macaques receiving the subtype E/C pentavalent vaccine. This is in contrast with the guinea pig study where binding to the C.TV1 V2.2 linear epitope was observed in guinea pigs receiving the subtype C pentavalent vaccine. This finding suggests that host factors (guinea pig vs. macaque) or the C.TV1 gp120 (in subtype C but not in the subtype E/C pentavalent vaccine) may have contributed to the V2.2 epitope response. In addition, our macaque study data showed that alum is as effective as, or superior to, the GLA-SE adjuvant in stimulating an antibody response that binds to V1V2 antigens and in blocking CH58 binding to Env. This is consistent with previous reports indicating the alum adjuvant as potentially advantageous for the development of protective antibody responses including those targeting V2 [47].

One caveat of the evaluation in this study, however, is that, unlike for the bivalent vaccine strains, vaccine-matched sequences for the trivalent stains were not available in the peptide microarray library. It is possible that the trivalent vaccine also induced anti-V2.hotspot response, and the anti-V2.2 response may not be limited to TV1, but our library did not allow for the detection of such responses due to the lack of matched sequences. The TV1 Env vaccine strain itself, on the other hand, did not elicit in guinea pigs the anti-V2.2 response as shown by results for the bivalent group. These data showed that the trivalent vaccine immunization was able to elicit an anti-V2.2 response that covers some heterologous strains. In addition, the V2 hotspot epitope is in close proximity to the V2.2 epitope. Antibodies specific for the V2 hotspot may still interfere with α4β7 and Env interaction. One of these antibodies, CH58, has been shown to partially block α4β7 binding [48], and a set of anti-linear V2 hotspot mAbs isolated from the RV144 extension study, RV305, also showed various levels of α4β7 blocking [49]. In addition, the smaller group sizes in both animal studies limit the power of the studies to reliably detect real, replicable differences among groups. Statistical analyses were performed to help identify trends for further explorations. Future investigations with larger group sizes are needed to confirm our findings and further characterize immunogenicity differences among different vaccine designs.

It is noteworthy that the three new Env strains were selected to complement the three subtype C strains, including C.96ZM651 which is included only in the prime for V1V2 breadth. Although this animal study included only the boost immunogens, without the C.96ZM651 prime immunogen, substantial enhancement of the V1V2 response breadth was still observed in the pentavalent group compared to the bivalent group. Based on the sequence analysis, the five strains in combination cover the most subtype C V1V2 variants, but not as completely as when all six strains are combined. The HVTN 705/IMBOKODO trial was a bivalent mosaic prime designed for global coverage followed by a C clade gp140 boost. This vaccine failed to show protection efficacy, though did show a trend in an association of greater IgG3 V1V2 antibody breadth with a lower likelihood of HIV acquisition [50]. These antigens were not specifically designed, however, to elicit a broad array of antibody specificities that targeted V1V2 within the C clade to mediate antibody Fc effector functions. Computational and novel targeting vaccine designs that target improving V1V2 breadth are important to test in further HIV-1 vaccine efficacy trials and to build upon the results from the current efficacy trials.

In summary, our data support the broadening of V1V2 binding antibody breadth with the inclusion of computationally selected subtype C Env strains in the pentavalent vaccine regimen. These results warrant further NHP and human clinical studies to test the efficacy of the pentavalent regimen and serve as a proof of concept of the merit of the computational immunogen design approach for enhancing the breadth of the immune response.

## Figures and Tables

**Figure 2 vaccines-13-00133-f002:**
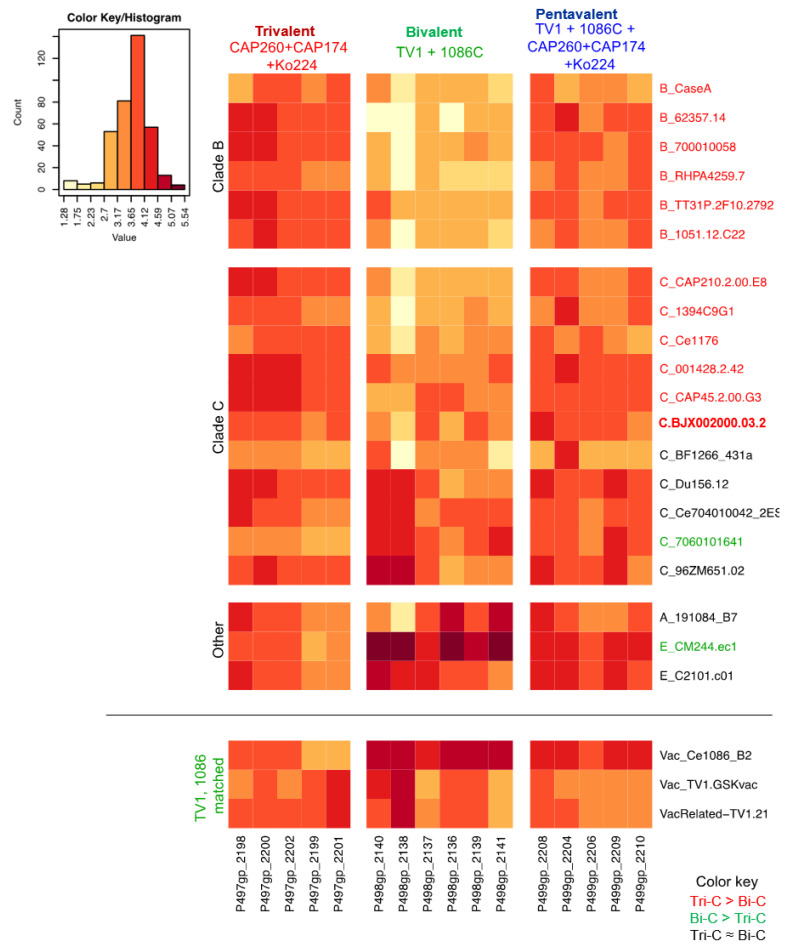
Heatmap of binding magnitude (EC50) for V1V2 gp70 scaffolds by guinea pigs in the 3 vaccine groups. V1V2 antigens in red showed higher binding (*p* < 0.05, two-tailed Wilcoxon rank sum test) by the trivalent group compared to the bivalent group, and green indicated higher binding by the bivalent group compared to the trivalent group. Antigens are ordered by subtypes. Antigens above the horizontal line are heterologous strains, and the ones below the line are vaccine-matched strains.

**Figure 3 vaccines-13-00133-f003:**
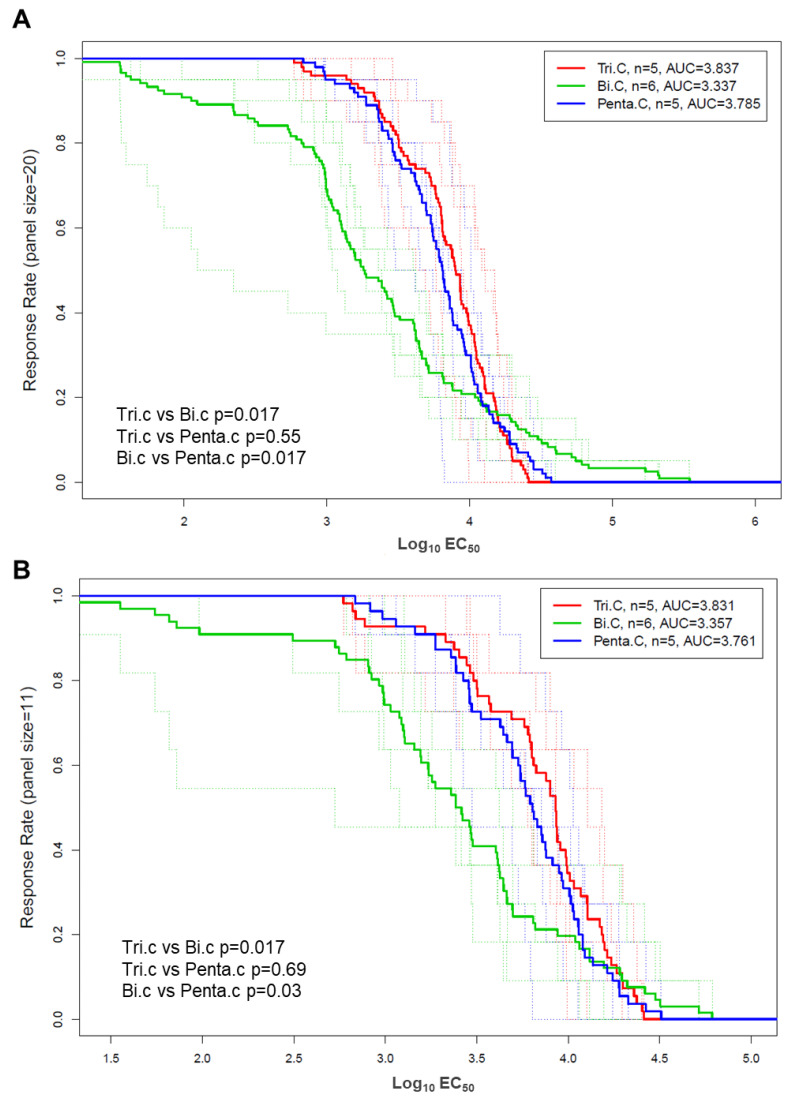
Magnitude–breadth curves for binding to 20 heterologous gp70 V1V2 (**A**) or 12 subtype C heterologous gp70 V1V2 scaffolded proteins (**B**) by guinea pigs in each group post 4th immunization. Each curve is plotted as the proportion of V1V2 antigens in the respective panel that the animal responded to with a magnitude (Log_10_ EC_50_) within the interval indicated on the x-axis. Thin dotted lines represent individual animals, and thick solid lines represent the group median values.

**Figure 4 vaccines-13-00133-f004:**
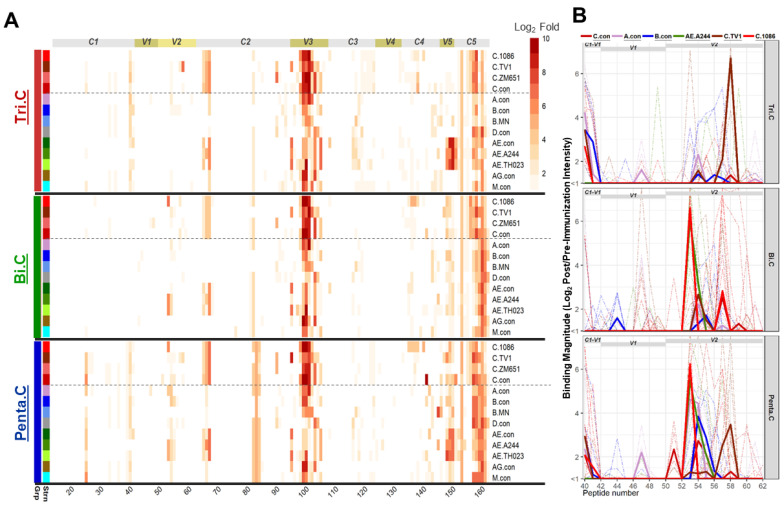
gp120 (**A**) and V1V2 region (**B**) linear epitope binding profiles for the 3 guinea pig vaccine groups. For the gp120 binding magnitude heatmap (**A**), each line represents a different HIV-1 Env strain in the array library, and values plotted are group median intensity values. The dotted lines separate subtype C strains from other strains. Relevant Env regions are labeled by bars on the top of the heatmap. For V1V2 binding profiles (**B**), thin dashed lines represent the binding magnitude of individual animals to virus strains indicated by color. Thick solid lines represent the group median binding magnitude for the respective strains. Regions within V1V2 are indicated as gray bars on top of the plots.

**Figure 5 vaccines-13-00133-f005:**
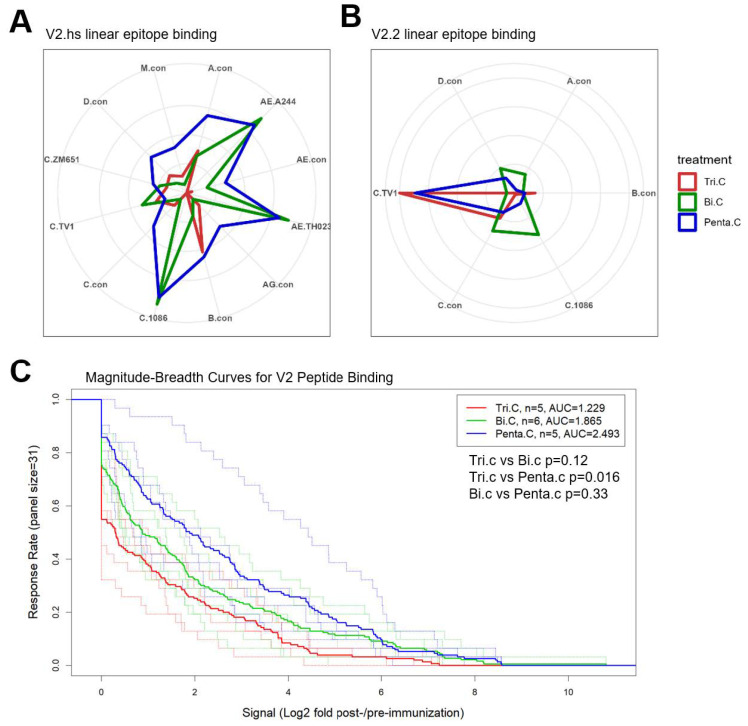
Spider plots showing the coverage Env strains for the two V2 linear epitopes by the 3 groups of guinea pigs (**A**,**B**), and the magnitude–breadth curves for binding to V2 linear peptides for the 3 vaccine groups. For the spider plots (**A**,**B**), labeled around each spider plot are the Env strains that showed responses for the respective epitope. The magnitude of binding to each epitope is the highest binding signal (Log_2_ fold post-/pre-immunization) to a single peptide within the epitope region for the strain. The epitope region covers peptide #53–54 for V2.hostpot, and peptide #57–58 for V2.2. Plotted are the group median magnitude values. For the magnitude–breadth curve plot (**C**), each curve is plotted as the proportion of linear V2 peptides (shown in Appendix A) that the animal responded to with a magnitude (Log_2_ fold post-/pre-immunization) within the interval indicated on the x-axis. Thin dotted lines represent individual animals, and thick solid lines represent the group median values.

**Figure 6 vaccines-13-00133-f006:**
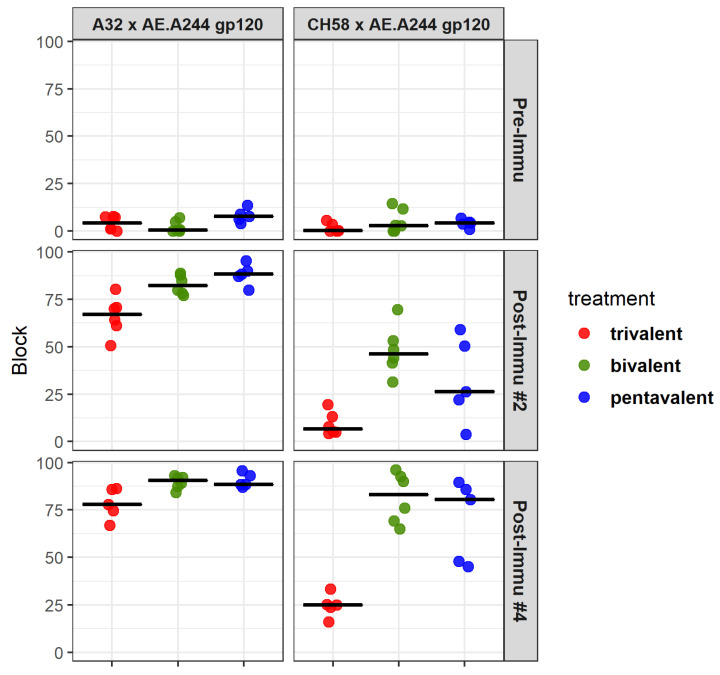
Magnitude of blocking by sera of immunized guinea pigs against binding of mAbs to Env antigens. The mAb by Env antigen pairs are indicated above each panel. Each symbol represents the response of one animal. Thick bars indicated the group median magnitude of blocking.

**Figure 7 vaccines-13-00133-f007:**
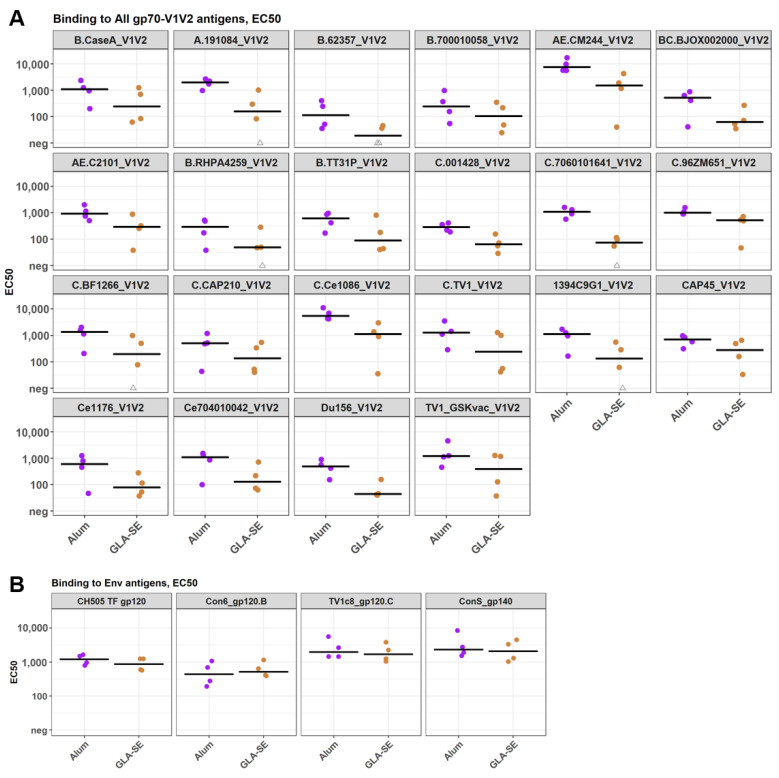
Serum IgG titers (EC_50_) of 2 pentavalent E/C immunized macaque groups measured in BAMA for binding to V1V2 gp70 scaffolds (**A**) and Env (**B**) antigens. Spots represent individual animals and are color-coded by group. Thick black bars represent the group median EC_50_.

**Figure 8 vaccines-13-00133-f008:**
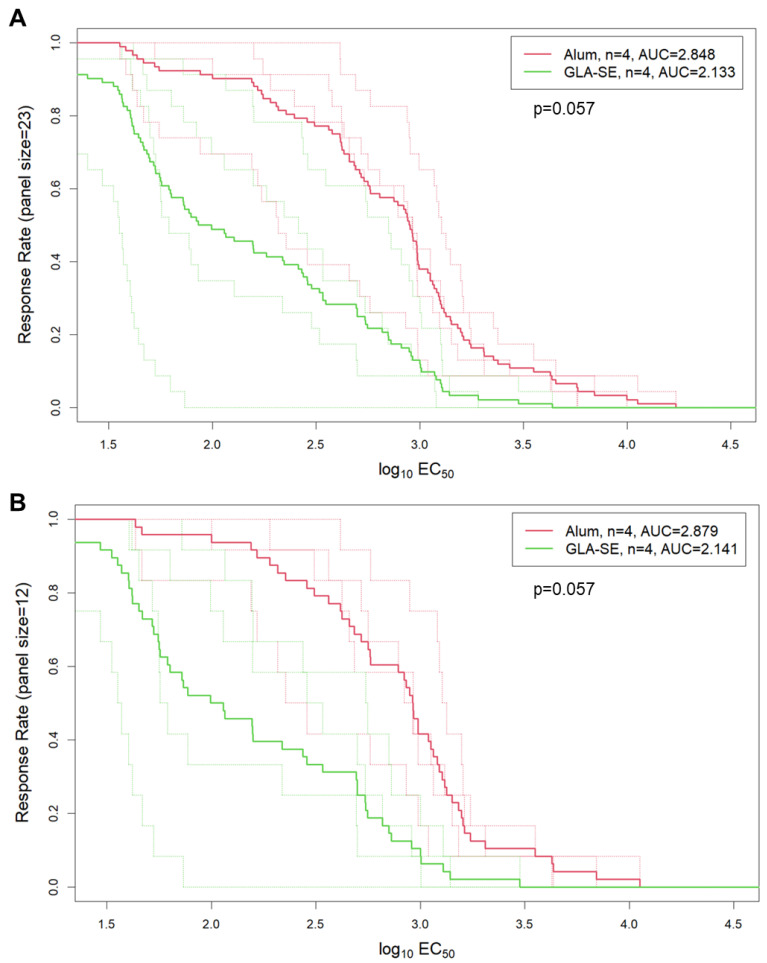
Magnitude–breadth curves for binding to all 23 gp70 V1V2 scaffolds (**A**) or 12 subtype C heterologous gp70 V1V2 (**B**) by 2 groups of pentavalent E/C immunized macaques at wk10. Each curve is plotted as the proportion of V1V2 antigens in the respective panel that the animal responded to with a magnitude (Log_10_ EC_50_) within the interval indicated on the x-axis. Thin dotted lines represent individual animals, and thick solid lines represent the group median values.

**Figure 9 vaccines-13-00133-f009:**
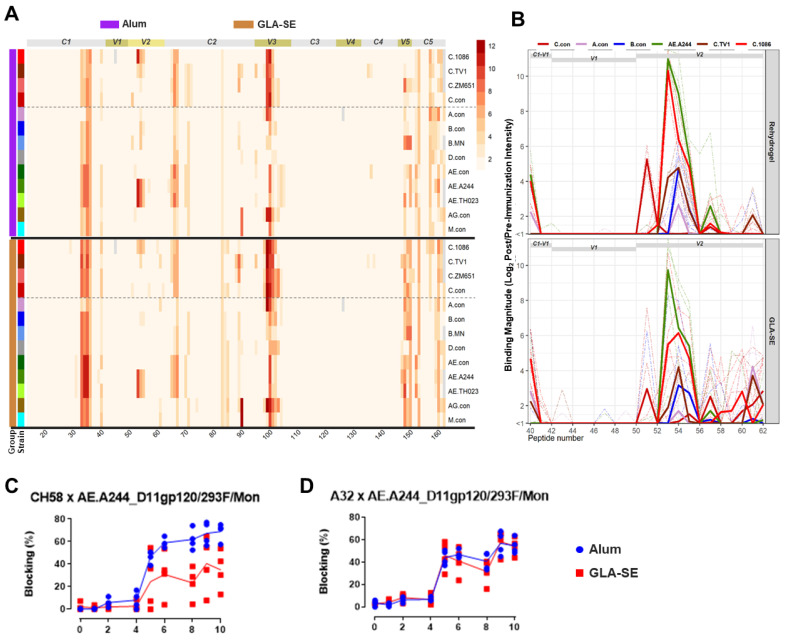
gp120 (**A**) and V1V2 region (**B**) linear epitope binding profiles for the 2 pentavalent E/C immunized macaque groups, and longitudinal blocking activity of serum antibodies to CH58 (**C**) and A32 (**D**) binding. For the gp120 binding magnitude heatmap (**A**), each line represents a different HIV-1 Env strain in the array library, and values plotted are group median intensity values. The dotted lines separate subtype C strains from other strains. Relevant Env regions are labeled by bars on the top of the heatmap. For V1V2 binding profiles (**B**), thin dashed lines represent the binding magnitude of individual animals to virus strains indicated by color. Thick solid lines represent the group median binding magnitude for the respective strains. Regions within V1V2 are indicated as gray bars on top of the plots.

**Table 1 vaccines-13-00133-t001:** Study design for the guinea pig immunization study.

	Trivalent (Tri-V)	Bivalent (Bi-V)	Pentavalent (Penta-V)
Immunogen	CAP260 gp120 Δ11 env CAP174 gp120 Δ11 env Ko224 gp120 Δ11 env	TV1c8 gp120 Δ11 env 1086C gp120 Δ7 env	CAP260 gp120 Δ11 env CAP174 gp120 Δ11 env Ko224 gp120 Δ11 env TV1c8 gp120 Δ11 env 1086C gp120 Δ7 env
Dose	100 μg per animal per protein	100 μg per animal per protein	100 μg per animal per protein
Adjuvant	STR8S-C	STR8S-C	STR8S-C
Immunization time	Days 0, 21, 42, 63	Days 0, 21, 42, 63	Days 0, 21, 42, 63
Group size	N = 5 *	N = 6	N = 5 *

* One animal each in the Tri-V and Penta-V group was euthanized after 4 immunizations due to weight loss. Both animals were found to have developed hepatic lipidosis upon necropsies. The pathologist could not determine with certainty that hepatic lipidosis was the result of the immunogen or adjuvant, but either potential cause could not be ruled out. Data for these 2 animals were not included in any of the data analyses.

**Table 2 vaccines-13-00133-t002:** Group median binding antibody titer (EC50) for V1V2 and Env antigen in BAMA and statistical test results for the comparison of binding magnitude among groups.

		Group Median			*p* Value *, Wilcoxon Rank Sum Test		
Category	Analyte	Tri-C	Bi-C	Penta-C	Tri- vs. Bi-C	Tri- vs. Penta-C	Bi- vs. Penta-C
Heterologous	gp70.001428.2.42_V1V2	19,671	2921	8903	** 0.004 **	0.310	** 0.004 **
V1V2	gp70.1051.12.C22_V1V2	9041	1046	2970	** 0.004 **	0.095	** 0.004 **
scaffold	gp70.1394C9G1_V1V2	7560	896	2897	** 0.009 **	0.841	** 0.004 **
	gp70.62357.14_V1V2	13,049	298	4633	** 0.004 **	0.151	** 0.004 **
	gp70.700010058_V1V2	10,312	989	6103	** 0.004 **	0.095	** 0.004 **
	gp70.B.CaseA_V1V2	5385	1228	1647	** 0.009 **	0.151	0.429
	gp70.BJOX002000.03.2_V1V2	9518	2542	5591	** 0.030 **	0.548	0.052
	gp70.CAP210.2.00.E8_V1V2	10,188	1105	5455	** 0.004 **	0.056	** 0.004 **
	gp70.CAP45.2.00.G3_V1V2	16,340	1998	10,624	** 0.004 **	0.222	** 0.009 **
	gp70.Ce1176_V1V2	7958	1702	4248	** 0.017 **	0.095	0.082
	gp70.RHPA4259.7_V1V2	6351	277	2301	** 0.004 **	0.310	** 0.004 **
	gp70.TT31P.2F10.2792_V1V2	12,452	1015	7609	** 0.004 **	0.421	** 0.009 **
	gp70.7060101641_V1V2	2790	14,347	11,982	** 0.017 **	0.056	0.329
	gp70.CM244.ec1_V1V2	6399	189,146	27,970	** 0.004 **	** 0.032 **	** 0.009 **
	gp70.191084_B7_V1V2	10,988	9728	10,177	0.662	1.000	0.662
	gp70.96ZM651.02_V1V2	9707	8667	6374	0.931	0.841	1.000
	gp70.BF1266_431a_V1V2	2141	3363	965	0.662	0.841	0.931
	gp70.C2101.c01_V1V2	11,113	14,661	14,489	0.429	0.310	0.931
	gp70.Ce704010042_2ES_V1V2	8530	10,108	9396	0.429	1.000	0.537
	gp70.Du156.12_V1V2	8725	3956	9215	0.429	1.000	0.537
Vaccine-	gp70.Ce1086_B2_V1V2	6338	63,496	29,731	** 0.004 **	** 0.008 **	** 0.004 **
matched	gp70.TV1.21_V1V2	8546	6296	4345	0.329	** 0.016 **	0.792
V1V2	gp70.TV1.GSKvacV1V2	5122	7338	2401	0.792	0.151	0.537
V1V2 and	AE.A244.V1V2_Tags	1530	6554	5594	** 0.009 **	0.056	0.429
V2 Tags	C.1086C.V1V2_Tags	1703	9758	5762	** 0.004 **	0.056	** 0.004 **
	AE.A244.V2_Tags	2488	5627	5137	0.126	0.222	0.537
	C.1086.V2_Tags	6135	4107	5065	0.537	0.690	0.329
Env	CH505.TF.gp120	44,770	33,091	40,088	0.247	0.548	0.662
	Con.6.gp120_B	17,953	9163	23,578	0.052	0.310	** 0.009 **
	Con.S.gp140.CFI	64,043	268,439	57,605	0.537	1.000	0.429
	TV1c8_D11gp120.avi	40,029	335,328	53,661	0.082	0.222	0.082

* Raw *p* values are not adjusted for multiple comparisons. *p* values < 0.05 are bolded, with colors to indicate which group was higher in the pair-wise comparison: red—trivalent group higher, green—bivalent group higher, and blue—pentavalent group higher. *p* values < 0.01 are bolded and underlined.

**Table 3 vaccines-13-00133-t003:** Group median binding intensity for V2 linear epitopes in peptide microarray and statistical test results for the comparison of binding magnitude among groups.

		Group Median			*p* Value *, Wilcoxon Rank Sum Test		
Epitope	Strain	Tri-C	Bi-C	Penta-C	Tri- vs. Bi-C	Tri- vs. Penta-C	Bi- vs. Penta-C
V2.hotspot	C.1086	0.0	6.6	6.2	** 0.004 **	** 0.008 **	0.628
V2.hotspot	C.TV1	1.8	2.7	1.3	0.429	0.802	0.502
V2.hotspot	AnyStrain ^#^	4.4	7.3	7.6	** 0.017 **	** 0.032 **	0.500
V2.2	C.1086	0.1	2.8	0.7	0.050	0.206	0.459
V2.2	C.TV1	6.7	0.5	5.8	** 0.022 **	0.841	0.082
V2.2	AnyStrain	6.7	6.6	6.0	1.000	0.841	0.931

* Raw *p* values are not adjusted for multiple comparisons. *p* values < 0.05 are bolded, with colors to indicate which group was higher in the pair-wise comparison: red—trivalent group higher, green—bivalent group higher, and blue—pentavalent group higher. *p* values < 0.01 are bolded and underlined. ^#^ AnyStrain is the highest binding to a single peptide in the epitope region (peptide #53–54 for V2.hotspot; peptide #57–58 for V2.2).

**Table 4 vaccines-13-00133-t004:** Comparison of specific antibody blocking activity among guinea pig immunization groups.

		Group Median			*p* Value *, Wilcoxon Rank Sum Test		
Time Point	Blocking Pair	Tri-C	Bi-C	Penta-C	Tri- vs. Bi-C	Tri- vs. Penta-C	Bi- vs. Penta-C
Post Immu.#2	A32 x AE.A244 gp120	67.1	82.3	88.3	** 0.016 **	** 0.011 **	0.1
Post Immu.#2	CH58 x AE.A244 gp120	6.6	46.3	26.4	** 0.004 **	0.100	0.27
Post Immu.#4	A32 x AE.A244 gp120	78.0	90.6	88.5	** 0.018 **	** 0.009 **	0.86
Post Immu.#4	CH58 x AE.A244 gp120	25.0	83.0	80.5	** 0.006 **	** 0.009 **	0.27

* Raw *p* values are not adjusted for multiple comparisons. *p* values < 0.05 are bolded, with colors to indicate which group was higher in the pair-wise comparison: red—trivalent group higher, green—bivalent group higher, and blue—pentavalent group higher. *p* values < 0.01 are bolded and underlined.

**Table 5 vaccines-13-00133-t005:** Study design for the cynomolgus macaque immunization study.

Group	Alum Pentavalent E/C	GLA-SE Pentavalent E/C
Immunogen	AE.A244_D11gp120/293F/Mon	AE.A244_D11gp120/293F/Mon
	CAP260 gp120 Δ11 env	CAP260 gp120 Δ11 env
	CAP174 gp120 Δ11 env	CAP174 gp120 Δ11 env
	Ko224 gp120 Δ11 env	Ko224 gp120 Δ11 env
	1086C gp120 Δ7 env	1086C gp120 Δ7 env
Dose	100 mg per animal per protein	100 mg per animal per protein
Adjuvant	Alum rehydrogel	GLA-SE
Immunization time	Week 0, 4, 8	Week 0, 4, 8
Group size	N = 4	N = 4

## Data Availability

The data presented in this study are available in this article and Appendix A. Raw data are available upon request.

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
