# Peer review of "A Pentavalent HIV-1 Subtype C Vaccine Containing Computationally Selected gp120 Strains Improves the Breadth of V1V2 Region Responses"

_vaccines, 2025, doi:10.3390/vaccines13020133_

Round 1
Reviewer 1 Report
Comments and Suggestions for Authors
Review of Manuscript “A Pentavalent HIV-1 Subtype C Vaccine Containing Computationally Selected gp120 Strains Improves the Breadth of V1V2 Region Responses “ by Xiaoying Shen et al..
Genetic diversity of different HIV-1 strains represents a major problem in developing an efficient vaccination strategy. In this very well written and very clearly structured manuscript the authors computationally selected three subtype C Env strains to maximize antibody binding coverage and epitope accessibility for complementation of a gp120 subtype C vaccine already used in human clinical trials in South Africa. These 3 novel subtype C Env strains were tested as a trivalent combination in a preclinical immunization study in guinea pigs, side by side with the original 1086C and TV1 bivalent regimen used as booster in the clinical trials. Additionally, a pentavalent regimen, which included all 5 subtype C strains, was tested. Both for the trivalent and the pentavalent groups, the overall breadth of the cross-subtype V1V2 binding as determined with V1V2 scaffolded proteins was clearly higher than that of the original bivalent vaccine. Whereas the bivalent vaccine efficiently elicited responses against the main V2 linear epitope (V2.hotspot) also targeted by the original RV144 vaccine in successful clinical trials in Thailand, the trivalent vaccine mainly elicited immune responses against a novel linear V2 epitope overlapping the alpha4beta7 binding motif (V2.2). The pentavalent vaccine elicited immune responses against both epitopes.
A slightly modified version of the pentavalent vaccine was also able to induce antibodies against a broad range of V1/V2 scaffolds in macaques. Interestingly, Alum was superior to GLA-SE as adjuvant in stimulating antibody response in the macaque model.
The well-presented data of the present study look quite promising and suggest further studies of this pentavalent regimen in non-human primates and in humans. Some points listed in detail below should be addressed in a revised version of the manuscript.
Major points:
1) With 6 animals per group in the guinea pig experiments, the study seems to be a bit underpowered, as judged from the variance in antibody titers (heatmaps in fig. 2). The authors should comment on this.
2) I could not quite follow the argumentation, why the ZM651 strain was not included in the control regimen (making it a trivalent vaccine). The authors should provide more details here.
Minor points:
1) The x-axis in figure 1C should probably read V1h and V2h length and not net charge.
2) Line 145/146: “the augmented diversity coverage of by trivalent shown in blue.“ should probably read “...by the trivalent vaccine…“.
3) Fig. 4A (and also S5): Is there a special reason that the usual color coding with green corresponding to the bivalent vaccine and red corresponding to the trivalent vaccine was changed in these figures (green now trivalent and red bivalent vaccine)?
4) Fig. 5A and B, again change in color coding (for bivalent and trivalent vaccine) as compared to the usual one and also different from fig. 5C.
5) Sentence in lines 581/582 seems to be incomplete (should probably be linked to the next sentence, replace dot by a comma).
Author Response
Comments: Genetic diversity of different HIV-1 strains represents a major problem in developing an efficient vaccination strategy. In this very well written and very clearly structured manuscript the authors computationally selected three subtype C Env strains to maximize antibody binding coverage and epitope accessibility for complementation of a gp120 subtype C vaccine already used in human clinical trials in South Africa. These 3 novel subtype C Env strains were tested as a trivalent combination in a preclinical immunization study in guinea pigs, side by side with the original 1086C and TV1 bivalent regimen used as booster in the clinical trials. Additionally, a pentavalent regimen, which included all 5 subtype C strains, was tested. Both for the trivalent and the pentavalent groups, the overall breadth of the cross-subtype V1V2 binding as determined with V1V2 scaffolded proteins was clearly higher than that of the original bivalent vaccine. Whereas the bivalent vaccine efficiently elicited responses against the main V2 linear epitope (V2.hotspot) also targeted by the original RV144 vaccine in successful clinical trials in Thailand, the trivalent vaccine mainly elicited immune responses against a novel linear V2 epitope overlapping the alpha4beta7 binding motif (V2.2). The pentavalent vaccine elicited immune responses against both epitopes.
A slightly modified version of the pentavalent vaccine was also able to induce antibodies against a broad range of V1/V2 scaffolds in macaques. Interestingly, Alum was superior to GLA-SE as adjuvant in stimulating antibody response in the macaque model.
The well-presented data of the present study look quite promising and suggest further studies of this pentavalent regimen in non-human primates and in humans. Some points listed in detail below should be addressed in a revised version of the manuscript.
Response: We thank the reviewer for the accurate summary and encouraging comments.
Major points:
Comments from Review 1
Genetic diversity of different HIV-1 strains represents a major problem in developing an efficient vaccination strategy. In this very well written and very clearly structured manuscript the authors computationally selected three subtype C Env strains to maximize antibody binding coverage and epitope accessibility for complementation of a gp120 subtype C vaccine already used in human clinical trials in South Africa. These 3 novel subtype C Env strains were tested as a trivalent combination in a preclinical immunization study in guinea pigs, side by side with the original 1086C and TV1 bivalent regimen used as booster in the clinical trials. Additionally, a pentavalent regimen, which included all 5 subtype C strains, was tested. Both for the trivalent and the pentavalent groups, the overall breadth of the cross-subtype V1V2 binding as determined with V1V2 scaffolded proteins was clearly higher than that of the original bivalent vaccine. Whereas the bivalent vaccine efficiently elicited responses against the main V2 linear epitope (V2.hotspot) also targeted by the original RV144 vaccine in successful clinical trials in Thailand, the trivalent vaccine mainly elicited immune responses against a novel linear V2 epitope overlapping the alpha4beta7 binding motif (V2.2). The pentavalent vaccine elicited immune responses against both epitopes.
A slightly modified version of the pentavalent vaccine was also able to induce antibodies against a broad range of V1/V2 scaffolds in macaques. Interestingly, Alum was superior to GLA-SE as adjuvant in stimulating antibody response in the macaque model.
The well-presented data of the present study look quite promising and suggest further studies of this pentavalent regimen in non-human primates and in humans. Some points listed in detail below should be addressed in a revised version of the manuscript.
Major points:
1) With 6 animals per group in the guinea pig experiments, the study seems to be a bit underpowered, as judged from the variance in antibody titers (heatmaps in fig. 2). The authors should comment on this.
Response: We agree with the reviewer on the limitation with the small group sizes. We have now included in Discussion: “In addition, the smaller group sizes in both animal studies limit the power of the studies to reliably detect real, replicable differences among groups…. Future investigations with larger group sizes are needed to confirm our findings and further characterize immunogenicity differences among different vaccine designs.” Lines 645-649.
2) I could not quite follow the argumentation, why the ZM651 strain was not included in the control regimen (making it a trivalent vaccine). The authors should provide more details here.
Response: ZM651 is the ALVAC prime of P5 clade C clinical studies. Our study is designed to improve the V1V2 coverage by the gp120 boost immunogens, assuming ZM651 would still be the ALVAC prime vaccine strain for an ALVAC-prime/gp120 boost vaccine regimen. Therefore, ZM651 is not included as a boost vaccine strain to test in the animal studies.
We’ve now included clarification in Results: “Here we only study the 2 boost strains from the P5 vaccine regimen and the newly selected subtype C Envs to optimize the boost Env strains. ZM651, which is the prime vaccine strain in ALVAC in the P5 studies, is not tested in these in the study because this study aimed to test for optimal boost strains. Diversity coverage by ZM651 would be provided by the ALAVAC prime in a prime/boost regimen using the optimal clade C Env protein combination determined by these exploratory studies as the boost vaccine. The impact of ZM651 on the diversity coverage is highlighted in Fig. 1”. Lines 278-285.
We also clarified in the legend for Fig.1: “ZM651, which was the assumed prime vaccine strain in the initial design”. Line 150.
Minor points:
1) The x-axis in figure 1C should probably read V1h and V2h length and not net charge.
Response: This error has now been corrected. We thank the reviewer for catching that.
2) Line 145/146: “the augmented diversity coverage of by trivalent shown in blue.“ should probably read “...by the trivalent vaccine…“.
Response: This sentence has now been corrected. Line 147.
3) Fig. 4A (and also S5): Is there a special reason that the usual color coding with green corresponding to the bivalent vaccine and red corresponding to the trivalent vaccine was changed in these figures (green now trivalent and red bivalent vaccine)?
Response: Colors for bivalent and trivalent groups have been switched in Fig. 4A and 5S to be consistent with other figures.
4) Fig. 5A and B, again change in color coding (for bivalent and trivalent vaccine) as compared to the usual one and also different from fig. 5C.
Response: Colors for bivalent and trivalent groups have been switched in Fig. 5A and B to be consistent with other figures.
5) Sentence in lines 581/582 seems to be incomplete (should probably be linked to the next sentence, replace dot by a comma).
Response: This sentence has now been corrected by removing “Whereas” at the beginning of the sentence. Line 601.
Reviewer 2 Report
Comments and Suggestions for Authors
Shen and colleagues used a computational approach to select HIV gp120 sequences for design of immunogens that aim to increase the breadth of antibody responses against V1V2 epitopes in the viral envelope. The authors tested the newly designed and previously used immunogens i) as a trivalent vaccine containing only the three selected novel subtype C gp120 peptides, ii) as bivalent vaccine comprising only HIV 1086C and TV1, 2 subtype C vaccine strains that were originally used as P5 subtype C boost strains, and iii) as a new pentavalent vaccine containing all 5 vaccine strains. The three immunogen combinations were first studied in guinea pigs and a variant of the pentavalent vaccine, consisting of the 4 subtype C strains in the pentavalent C vaccine combined with the AE.A244 strain, was also tested in macaques. An increased breadth of V1V2 targeting antibody response is presumed to improve immune protection against HIV infection. Breadth of V1V2 responses increased with the trivalent vaccine compared to immunization with the bivalent 1086C and TV1 based immunogens. Compared to both the bi- and trivalent vaccines the pentavalent combination elicited the largest breadth of antibody responses against a V1V2 scaffold panel and gp120 variants. The antibody responses were also evaluated in terms of their ability to compete with two antibodies that target epitopes linked to protection. In the macaque experiments, the authors also find that Alum hydrogel is more effective than a glucopyranosyl lipid adjuvant in a squalene-in-water emulsion (GLA-SE). Given the continuation of the HIV epidemic and urgent need for a protective vaccine, this exploratory study is timely and interesting. The authors also acknowledge limitations. However, there are a few points that need clarification:
Lines 148-149 and 271-273, if the most common missed amino acids would have been covered by the inclusion of ZM651, why was it not included in this study? Including most, but not all possible, major subtype C amino acid variants that were not covered by the original P5 boost strains seems a sub-optimal choice and the rationale for it is unclear.
Line 159, Section 2.2, 200µl per intramuscular injection side seems extremely high for guinea pigs. Also, lines 167-168 state ‘animals were immunized 4 times on Days 0, 21, 42, and 63 of the study’ and line 278 states immunization ‘4 times a week 0, 3, 6 and 9’. Just to be sure, were the animals immunized a total of 4 times, once each on day 0, 21, 42, and 63, or 4 times each on the 4 time points? The approach with 4 times total, once at a given day, seems what the authors mean to say. There is also some information about the animals missing: i) What was the sex of the animals? ii) What were the ages of the animals when the experiments were started? iii) Which were the suppliers of the animals? iv) Beyond suppliers, what is the origin of the animals or strain? Guinea pigs are often outbred, and there are various geographical locations that are sources for different rhesus macaque lines and colonies. All that information is important since all those parameter affect immune responses.
Lines 226-227, states “… No statistical comparisons were performed for the NHP study due to the small group size of the study.” The small group size is no reason to omit statistical analysis. Even if a significance level of p ≤ 0.05 is not reached, any trend may be informative.
Lines 206-207, the sentence contains word duplications: ‘was included as a negative was included as a negative’.
Table 1, the * in the last row is not explained but seems to label the groups in which animals died during the experiment. Were there any additional adverse reactions by the animals to the adjuvant or peptides/proteins.
Most references are missing the journal names. Reference 1 should be updated to a version more recent than 2019.
Comments on the Quality of English LanguageSome English language editing will be necessary.
Author Response
Comments from Review 2
Shen and colleagues used a computational approach to select HIV gp120 sequences for design of immunogens that aim to increase the breadth of antibody responses against V1V2 epitopes in the viral envelope. The authors tested the newly designed and previously used immunogens i) as a trivalent vaccine containing only the three selected novel subtype C gp120 peptides, ii) as bivalent vaccine comprising only HIV 1086C and TV1, 2 subtype C vaccine strains that were originally used as P5 subtype C boost strains, and iii) as a new pentavalent vaccine containing all 5 vaccine strains. The three immunogen combinations were first studied in guinea pigs and a variant of the pentavalent vaccine, consisting of the 4 subtype C strains in the pentavalent C vaccine combined with the AE.A244 strain, was also tested in macaques. An increased breadth of V1V2 targeting antibody response is presumed to improve immune protection against HIV infection. Breadth of V1V2 responses increased with the trivalent vaccine compared to immunization with the bivalent 1086C and TV1 based immunogens. Compared to both the bi- and trivalent vaccines the pentavalent combination elicited the largest breadth of antibody responses against a V1V2 scaffold panel and gp120 variants. The antibody responses were also evaluated in terms of their ability to compete with two antibodies that target epitopes linked to protection. In the macaque experiments, the authors also find that Alum hydrogel is more effective than a glucopyranosyl lipid adjuvant in a squalene-in-water emulsion (GLA-SE). Given the continuation of the HIV epidemic and urgent need for a protective vaccine, this exploratory study is timely and interesting. The authors also acknowledge limitations. However, there are a few points that need clarification:
Lines 148-149 and 271-273, if the most common missed amino acids would have been covered by the inclusion of ZM651, why was it not included in this study? Including most, but not all possible, major subtype C amino acid variants that were not covered by the original P5 boost strains seems a sub-optimal choice and the rationale for it is unclear.
ZM651 is the ALVAC prime of P5 clade C clinical studies. Our study is designed to improve the V1V2 coverage by the gp120 boost immunogens, assuming ZM651 would still be in the ALVAC prime for an ALVAC-prime/gp120 boost vaccine regimen. Therefore ZM651 is not included as a boost vaccine strain.
Response: ZM651 is the ALVAC prime of P5 clade C clinical studies. Our study is designed to improve the V1V2 coverage by the gp120 boost immunogens, assuming ZM651 would still be the ALVAC prime vaccine strain for an ALVAC-prime/gp120 boost vaccine regimen. Therefore, ZM651 is not included as a boost vaccine strain to test in the animal studies.
We’ve now included clarification in Results: “Here we only study the 2 boost strains from the P5 vaccine regimen and the newly selected subtype C Envs to optimize the boost Env strains. ZM651, which is the prime vaccine strain in ALVAC in the P5 studies, is not tested in these in the study because this study aimed to test for optimal boost strains. Diversity coverage by ZM651 would be provided by the ALAVAC prime in a prime/boost regimen using the optimal clade C Env protein combination determined by these exploratory studies as the boost vaccine. The impact of ZM651 on the diversity coverage is highlighted in Fig. 1”. Lines 278-285.
We also clarified in the legend for Fig.1: “ZM651, which was the assumed prime vaccine strain in the initial design”. Line 150.
Line 159, Section 2.2, 200µl per intramuscular injection side seems extremely high for guinea pigs. Also, lines 167-168 state ‘animals were immunized 4 times on Days 0, 21, 42, and 63 of the study’ and line 278 states immunization ‘4 times a week 0, 3, 6 and 9’. Just to be sure, were the animals immunized a total of 4 times, once each on day 0, 21, 42, and 63, or 4 times each on the 4 time points? The approach with 4 times total, once at a given day, seems what the authors mean to say. There is also some information about the animals missing: i) What was the sex of the animals? ii) What were the ages of the animals when the experiments were started? iii) Which were the suppliers of the animals? iv) Beyond suppliers, what is the origin of the animals or strain? Guinea pigs are often outbred, and there are various geographical locations that are sources for different rhesus macaque lines and colonies. All that information is important since all those parameter affect immune responses.
Response: 200ul x 2 sites IM has been the standard dosing regimen for guinea pigs for 30+ years in the animal facility. It is approved by our IACUC and is tolerated extremely well by the animals, with no lameness or ambulatory issues.
The animals were immunized on wk 0, 3, 6 and 9 for 4 immunizations total. We have now clarified/corrected at both places: “All animals were immunized on week 0, 3, 6, and 9 of the study for a total of 4 immunizations.” Lines 173-174. “…delivered 4 times total on week 0, 3, 6, and 9”. Line 290.
- i) All animals were female, and ii) were 3-4 months of age at the time of the first immunizations. Animals were iii) provided by Bioqual and were iv) Hartley outbred strain from Charles River.
We have now provided additional information on the animals in both studies in the manuscript. “Guinea pigs in the study were female, Hartley outbred strain from Charles Reiver, and were 3-4 months of age at the time of the first immunization.” Lines 163-165. “Mauritian cynomolgus macaques for the NHP study…” and “6 adult female macaques and 2 adult male macaques were split into 2 groups with one male per group. Each macaque received 500 ug total Env protein (100 ug each Env), adjuvanted with either alum/rehydrogel or GLA-SE (25 μg) in a total of 1mL and split into 0.5 mL per side (bilateral quadriceps).” Line 175 and Lines 175-179.
Lines 226-227, states “… No statistical comparisons were performed for the NHP study due to the small group size of the study.” The small group size is no reason to omit statistical analysis. Even if a significance level of p ≤ 0.05 is not reached, any trend may be informative.
Response: We have included statistical analysis for the NHP study as the reviewer recommended. Statists are indicated on Fig. 8 and summarized in Table S1 (newly added). References for the statistics and Table S1 are added in Results. Lines 528 and 548-549.
We also included in our discussion “Statistical analyses were done to help identify trends for further explorations” to clarify the purpose of the analyses. Lines 647-648.
Lines 206-207, the sentence contains word duplications: ‘was included as a negative was included as a negative’.
Response: The repeated “was included as a negative” is now removed. Line 214.
Table 1, the * in the last row is not explained but seems to label the groups in which animals died during the experiment. Were there any additional adverse reactions by the animals to the adjuvant or peptides/proteins.
Response: Yes * indicates animal death in the group. We have now added a table footnote: “*One animal each in the Tri-V and Penta-V group was euthanized after 4 immunizations due to weight loss. Both animals were found to have developed hepatic lipidosis upon necropsies. The pathologist could not determine with certainty that hepatic lipidosis was the result of the immunogen or adjuvant, but either potential cause could not be ruled out. Data for these 2 animals were not included in any of the data analyses.” Lines 305-309.
We thank the reviewer for catching the missing footnote.
Most references are missing the journal names. Reference 1 should be updated to a version more recent than 2019.
Response: References have now been reformatted to display journal names correctly. We thank the reviewer for catching the mistake!
Reference 1 (currently reference #13) has now been updated with correct report year. Numbers cited in paper were from 2023 statistics (2024 report) thus require no update.
-----------
In addition to response to reviewers’ comments, we made minor linguistic edits and corrected a few typos throughout the manuscript and provided updated Figures with large text “Fig. X” on top of each figure in previous version removed.